# Differential Scanning Calorimetry Analyses of Idebenone-Loaded Solid Lipid Nanoparticles Interactions with a Model of Bio-Membrane: A Comparison with In Vitro Skin Permeation Data

**DOI:** 10.3390/ph11040138

**Published:** 2018-12-16

**Authors:** Lucia Montenegro, Francesco Castelli, Maria Grazia Sarpietro

**Affiliations:** Department of Drug Sciences, University of Catania, V.le A. Doria 6, 95125 Catania, Italy; fcastelli@unict.it (F.C.); mg.sarpietro@unict.it (M.G.S.)

**Keywords:** differential scanning calorimetry, idebenone, solid lipid nanoparticles, in vitro, skin penetration, bio-membrane interactions

## Abstract

Differential scanning calorimetry (DSC) has emerged as a helpful technique both to characterize drug delivery systems and to study their interactions with bio-membranes. In this work, we compared idebenone (IDE)-loaded solid lipid nanoparticle (SLN) interactions with bio-membranes assessed by DSC with previous in vitro skin penetration data to evaluate the feasibility of predicting IDE skin penetration using DSC analyses. In vitro interactions experiments were performed using multi-lamellar liposomes as a model of bio-membrane. Enthalpy changes (ΔH) and transition temperature (Tm) were assessed during nine repeated DSC scans to evaluate IDE-loaded SLN–bio-membrane interactions over time. Analyzing ΔH and Tm values for each scan, we observed that the difference of ΔH and Tm values between the first and the last scan seemed to be related to SLN ability to locate IDE in the epidermis and in the stratum corneum, respectively. Therefore, the results of this study suggest the possibility of qualitatively predicting in vitro IDE skin penetration from IDE-loaded SLN utilizing the calorimetric parameters obtained from interaction experiments between the carriers under investigation and a model of bio-membrane.

## 1. Introduction

The ability to interact with biological membranes is one of the fundamental requisites for drugs to be effective. At membrane level, these interactions may involve both protein and lipid domains, as a drug could bind to proteins while perturbing the lipid phase, thus inducing membrane changes, which in turn could affect protein binding ability [1,2]. As drug–lipid interactions may play a crucial role in determining drug effectiveness, many studies have been performed on lipid-based systems as models of bio-membranes [3,4]. Among different lipid arrangements, liposomes, consisting of lipid bilayers organized to form vesicles, have drawn a great deal of attention [5]. Due to their structural analogy with the highly organized aqueous and lipid domains of the stratum corneum, the outermost layer of the skin representing the main barrier to drug skin permeation, liposomes have been proposed as model of skin membrane to evaluate drug interactions with the skin [6]. 

In recent years, differential scanning calorimetry (DSC) has emerged as a very helpful technique in the pharmaceutical field to characterize drug delivery systems and to investigate the interactions between nanocarriers and drugs as well as nanocarriers and bio-membranes [7,8,9,10,11].

In previous studies, we have prepared and characterized solid lipid nanoparticles (SLN) as carriers to improve topical bioavailability of idebenone (IDE), a synthetic analogue of coenzyme Q_10_ with strong antioxidant activity [12]. In vitro skin permeation studies of these SLN showed that IDE penetration into the different skin layers depended on the SLN composition, pointing out an IDE targeting of the upper skin layers. DSC analyses of the interaction between these IDE-loaded SLN and multi-lamellar (MLV) liposomes as models of bio-membranes revealed that IDE ability to penetrate into the liposome bilayers increased when IDE was loaded into SLN [13].

In this work, we have investigated IDE-loaded SLN interactions with a model of bio-membranes consisting of MLV liposomes performing repeated DSC scans over time and we have compared the resulting calorimetric data with previous in vitro skin penetration data to evaluate the feasibility of predicting IDE skin penetration from SLN using DSC analyses. 

The possibility of predicting skin penetration of drugs loaded into SLN using a simple instrumental technique like DSC would allow a fast screening of the most suitable carriers while reducing the number of expensive and time-consuming in vitro skin permeation experiments. 

## 2. Results and Discussion

### 2.1. SLN Characterization

Unloaded and IDE-loaded SLN showed morphological and physic-chemical properties similar to those previously reported [12]. Roughly round-shaped nanoparticles (Figure 1) had mean sizes in the range 34–47 nm, narrow dimensional distribution (polydispersity index <0.300) and slightly negative ζ potential (Table 1). Despite their low ζ potential values, unloaded and IDE-loaded SLN were stable after storage at room temperature for at least two months, as previously reported [12]. The high phase inversion temperature (PIT) values of these SLN supported their good stability. 

IDE loading capacity was determined as the maximum amount of IDE that could be incorporated in the nanoparticles giving rise to a clear formulation. Due to IDE poor water solubility, in a clear vehicle, IDE had to be completely incorporated into the SLN; otherwise, it would produce a turbid system and/or a precipitate. Other authors have used the same method to determine SLN loading capacity for other lipophilic drugs [14]. SLN A, prepared using isoceteth-20 as surfactant, showed a lower loading capacity (0.7% *w/w*) compared to SLN B and C (1.1% *w/w*), which could be attributed to the steric hindrance of the branched acyl chain in the surfactant structure, which could prevent a higher drug loading.

As shown in Table 1, a particle size decrease was observed loading 1.1% *w/w* IDE into SLN B while SLN C particle size was not affected by the amount of IDE loaded. In a previous study [12], this behavior has been attributed to interactions between IDE and surfactant layer, mainly depending on the different chemical structure of the surfactants used to prepare these SLN (isoceteth-20: branched acyl chain; ceteth-20: saturated linear acyl chain; oleth-20: unsaturated linear acyl chain).

### 2.2. IDE-Loaded SLN Interactions with MLV Liposomes

DSC analysis was performed to evaluate the interaction of SLN with multi-lamellar (MLV) liposomes, a model of cell membrane. Liposomes are a simplified system that effectively mimics many biological membrane properties [15]. When subjected to heat treatment, phospholipid bilayers undergo a phase transition from a gel-ordered state (Lβ), where phospholipids are tightly packed, to a liquid-crystalline state (Lα), in which phospholipids are more loosely associated [16]. This phase transition is detectable as an endothermic peak using DSC. In the case of 1,2-dimyristoyl-sn-glycero-3-phosphatidylcholine (DMPC), the phase transition occurs at about 25 °C, and a small peak is also recorded at about 15 °C. The latter is related to the transition from the gel to the ripple phase in which the bilayer surface is undulated and the phospholipid chains are tilted with respect to the normal layer [17].

The presence of foreign molecules in the phospholipid bilayer affects the van der Waals forces among the acyl chains and the interactions among the polar heads, resulting in the variation of thermodynamic parameters such as transition temperature (T_m_) and enthalpy change (ΔH).

Previous DSC studies highlighted that the use of the phase inversion temperature (PIT) method allowed obtaining SLN as their thermograms showed a main T_m_ about 12 °C lower than that of the cetyl palmitate bulk, thus indicating that cetyl palmitate (located in the SLN core) was solidified [18,19].

In a recent study, DSC characterization of these SLN showed the influence of surfactants on IDE distribution within the nanoparticles [19,20]. Therefore, in this paper we studied the interactions of IDE-loaded SLN with a membrane model and the surfactant effects on these interactions.

Before evaluating the interaction between DMPC MLV and SLN, we analyzed each sample of unloaded SLN, MLV of pure DMPC and MLV loaded with IDE at molar fraction 0.06, to obtain the reference curves. Then, the interaction study was carried out. These calorimetric experiments were performed leaving SLN in contact with MLV at 37 °C to mimic the physiological temperature. In addition, unloaded SLN were analyzed to evaluate if the carrier alone could interact with the bio-membrane. The interaction was evaluated by heating scans carried out every hour starting from the first that was performed immediately after putting the SLN sample in contact with MLV.

Thermograms of unloaded SLN put in contact with MLV are shown in Figure 2. The obtained curves were compared with those of pure DMPC MLV and unloaded SLN. With regard to SLN A (Figure 2A), in the first scan the pre-transition and main peak of MLV and the SLN peak can be recognized. From the second scan, the pre-transition peak gradually disappeared; the main peak became sharper and decreased in intensity. SLN peak did not undergo noticeable changes. These data indicated that SLN A were able to enter into the DMPC bilayers but retained their integrity as the corresponding unchanged peak indicated. Figure 2B shows the thermograms of unloaded SLN B. The first scan indicated that MLV and SLN were separate entities that gave rise to peaks superimposable to those of the references. From the second scan (after one hour of contact), the SLN entered into the phospholipid bilayers causing the disappearance of the pre-transition peak and variations of both the shape and intensity of the MLV main peak. SLN B peak remained unchanged during all the experiments indicating that the nanoparticle structure was not altered. Thermograms of unloaded SLN C and MLV (Figure 2C) highlighted that these nanoparticles deeply affected the thermotropic behavior of DMPC liposomes. As soon as put in contact (first scan), MLV and SLN did not seem to interact (the thermograms of MLV and SLN C can be superimposed), but deep variations occurred over time. In particular, in the second scan, the disappearance of the pre-transition peak could be observed, along with the presence of a shoulder before the MLV main peak. In the successive scans, the shoulder and the MLV main peak gradually merged in a unique peak that shifted toward lower temperature and decreased in intensity. SLN C peak remained almost unchanged. These data revealed that SLN C entered into MLV; at the beginning, they distributed non-homogeneously, being located mainly in the outer bilayers of MLV (as the presence of the shoulder indicated); subsequently, they moved toward the inner layers of MLV (shoulder and main peak merging). When crossing the bilayers, SLN C maintained their structural integrity.

With regard to IDE-loaded SLN thermograms, they were compared with calorimetric curves of pure DMPC MLV, MLV loaded with 0.06 molar fraction of IDE and the corresponding unloaded SLN. In Figure 3, loaded SLN A/MLV thermograms are shown. In both SLN A1 and SLN A2 thermograms, the pre-transition peak of MLV disappeared while the MLV main peak became narrower and moved toward lower temperatures. The small peak at higher temperature of loaded SLN A, which was due to IDE presence [13], disappeared immediately indicating a fast release of IDE to the bio-membrane. The peak of IDE loaded SLN was similar to that of unloaded SLN. The data clearly suggested a fast release of IDE from SLN to MLV. With regard to the peak of SLN, in SLN A1 and SLN A2, the shoulder at high temperature was lost. 

Similarly, IDE-loaded SLN B related thermograms (Figure 4) show the disappearance of the pre-transition peak and the shrinkage of the main peak of MLV. Concerning IDE loaded SLN C thermograms (Figure 5), in the first scan of SLN C1 and SLN C2 the pre-transition peak decreased, the MLV main peak became sharper and a unique peak replaced the SLN peak. In the second scan, the pre-transition peak disappeared and the main peak split into two signals. In the successive scans, the two signals related to MLV peak gradually merged into a single peak that became smaller and smaller and moved toward lower temperatures. In the first scan of SLN C3, we observed a pre-transition peak, a main peak related to MLV and a broad peak that replaced the two peaks of SLN, localized at a higher temperature with respect to loaded SLN C3. From the second scan, the pre-transition disappeared whereas a shoulder and a peak replaced the main peak. In the successive scans, shoulder and peak gave rise to a broader peak that moved toward a lower temperature approaching that of IDE-loaded MLV. Therefore, the kinetic calorimetric curves of SLN C1 and SLN C2 were very similar to the corresponding curves of unloaded SLN C. On the contrary, SLN C3 thermograms (Figure 5C) exhibited differences in the MLV peak. In the second scan of SLN C3 thermograms, the pre-transition peak was not observed and the MLV peak split into two broad peaks, which gradually merged into a unique very wide peak as the experiment proceeded. In the last scan, the peak temperature was very close to that of IDE-loaded MLV. The different behavior observed among SLN C3 and other SLN C could be ascribed to the highest relative amount of IDE with respect to SLN. In fact, in all experiments the amount of SLN put in contact with MLV was calculated in order to have 0.06 molar fraction of IDE with respect to DMPC. Accordingly, the total amount of IDE in each experiment was the same in all three cases (SLN C1, C2 and C3). As IDE loading in the three SLN C was different, the total amount of SLN was different and decreased from SLN C1 to SLN C3. In SLN C1 and SLN C2 thermograms, the signal due to the interaction of SLN with MLV prevailed, hiding the effect of IDE. Conversely, in SLN C3 thermograms the interaction of IDE with MLV was visible, as the peak temperature value indicated, and the interaction of SLN with MLV caused the widening of the peak. 

Summarizing the results obtained, all tested SLN were able to insert immediately in MLV bilayers and to release IDE. SLN A and SLN B (containing isoceteth-20 and ceteth-20, respectively, as surfactant) showed a different interaction compared to SLN C (containing oleth-20 as surfactant). In fact, SLN C modified more deeply the thermotropic behavior of MLV than SLN A and SLN B. SLN C3 caused the clearest changes on the MLV peak, which probably were due to the contribution of IDE. 

Although the concentration ratio between IDE and SLN components was similar in SLN B3 and SLN C3, these SLN showed a different behavior because in SLN B3 thermograms, IDE contribution was not observed. This difference was surely due to the type of surfactant used in SLN B and SLN C, which was the only difference between these SLN. In a recent work [21], a different arrangement of IDE within the nanoparticles was postulated depending on the kind of surfactant. In particular, IDE was located in the SLN core when a saturated surfactant was used. On the other hand, the unsaturated surfactant caused the localization of IDE mainly at the nanoparticle surface. This IDE positioning could explain the different influence of SLN B3 and SLN C3 on the thermotropic behavior of MLV. In fact, being located more externally in SLN C3, IDE release from these SLN was probably greater than from SLN B3. 

### 2.3. Comparison between DSC and In Vitro Skin Penetration Data

The enhancement of skin penetration of drugs loaded into SLN has been attributed to different factors. Topical application of SLN forms an occlusive film on the skin surface, thus increasing skin hydration and hence drug penetration [20,22,23]. However, this occlusive effect depends on different factors, such as nanoparticles’ degree of crystallinity, size and number. Some investigations have pointed out that the higher the degree of crystallinity, the greater the occlusive effect [24,25]. Reducing particle size and/or increasing their number in the formulation leads to an enhancement of occlusion, as well [26]. In addition, the use of SLN as topical carriers shows several advantages, including prevention of drug systemic absorption and related side effects, drug targeting to specific skin layers, and good tolerability as SLN can be applied on damaged or inflamed skin owing to their non-irritant and non-toxic ingredients [27,28,29,30]. 

Schlupp et al. [31] investigated skin penetration of three glucocorticoids from SLN, concluding that lipid nature and nano-size of SLN could be fundamental in determining specific drug-carrier interactions with the skin that could lead to an enhancement of drug skin penetration.

Our DSC data outlined that the type of surfactant and the amount of drug loaded into the SLN could be further relevant parameters involved in drug–SLN interactions with the skin.

Previous in vitro skin permeation studies, on IDE-loaded SLN with the same composition, pointed out that the amount of drug penetrating different skin layers depended on the surfactant used to obtain SLN and on the amount of IDE loaded in these SLN [12]. As shown in Table 2, SLN containing oleth-20 as surfactant provided the highest drug penetration in the stratum corneum when loading 1.1% IDE (SLN C3). 

Accordingly, our DSC study indicates a stronger interaction of SLN C3 with a bio-membrane model with respect to other investigated SLN.

Therefore, in this work we assessed the possibility of correlating previous in vitro skin penetration results observed for IDE-loaded SLN with DSC data obtained studying the interactions of these SLN with a model of bio-membrane consisting of MLV liposomes. 

In Table 3 and Table 4, we report enthalpy changes (ΔH) and T_m_ values, respectively, determined during in vitro experiments on IDE-loaded SLN interactions with MLV liposomes. As shown in Table 3, ΔH values decreased from scan 1 to scan 4 for each SLN sample. However, approximately after 7–8 scans, this parameter did not show significant changes. Plotting ΔH values determined after each scan against the cumulative amount of IDE penetrating the different skin layers (stratum corneum, epidermis and dermis) from IDE-loaded SLN, we did not observe any relationship (*R^2^* < 0.3 for all comparisons, data not shown). The data reported in Table 3 show that in the first scan, ΔH values obtained for the tested SLN were very close (range 27–30 J) while in the last scan (scan 9) this parameter showed a broader range of values (26–15 J). Therefore, we thought it was noteworthy to calculate the difference between ΔH value obtained in scan 1 and scan 9 (ΔH1–9) for each SLN sample (see Table 3). Plotting ΔH1–9 values obtained for each IDE-loaded SLN against the corresponding cumulative amount penetrating the different skin layers, we pointed out a good correlation (*R^2^* = 0.925) between ΔH1–9 values and the IDE amount in the epidermis (Figure 6). 

T_m_ values showed very slight decreases from scan 1 to scan 9 for all IDE-loaded SLN, apart for SLN C3, whose interactions with MLV liposomes significantly lowered T_m_ value during the course of the experiments. 

Plotting T_m_ values, determined after each scan, against the cumulative amount of IDE penetrated in the stratum corneum from IDE-loaded SLN, we observed an almost linear relationship for all comparisons (*R^2^* ranging from 0.606 to 0.626), apart for scan 1, for which no correlation was found between the data under investigation (*R^2^* = 0.055). In Figure 7, we report, as an example, the plot of T_m_ values recorded during the last scan (scan 9) vs. the cumulative amount of IDE penetrated in the stratum corneum for each IDE-loaded SLN. As described for ΔH values, we calculated the difference of T_m_ between scan1 and scan 9 (ΔT_m_ 1–9) for each SLN sample. Plotting ΔT_m_ 1–9 values vs. the cumulative amount of IDE penetrated in the stratum corneum, we evidenced an almost linear correlation (*R^2^* = 0.704, Figure 8). 

We did not observe any relationship between T_m_1–9 values and the cumulative amount of IDE penetrating the epidermis or in the dermis. 

These results suggest that proper and in-depth analyses of DSC data obtained from in vitro interactions experiments using a model of bio-membrane could provide useful information to predict the amount of drug that could penetrate different skin layers from different SLN after in vitro topical application. In particular, ΔH1–9 and ΔT_m_1–9 seems to be related to SLN ability to locate IDE in the epidermis and in the stratum corneum, respectively. 

## 3. Materials and Methods

### 3.1. Materials

1,2-Dimyristoyl-sn-glycero-3-phosphatidylcholine (DMPC) (purity = 99%) was supplied by Genzyme Pharmaceuticals (Liestal, Switzerland). Methanol and chloroform were of LC grade and were bought from VWR (Darmstadt, Germany). Tris(hydroxymethyl)-aminomethane (Tris) was purchased from Merck (Germany). A 50 mM Tris solution, at pH 7.4, was used to prepare MLV. Fluka (Milan, Italy) supplied Polyoxyethylene-20-cetyl ether (Ceteth-20). Polyoxyethylene-20-isohexadecyl ether (Isoceteth-20) was a kind gift of Bregaglio (Milan, Italy). Polyoxyethylene-20-oleyl ether (Oleth-20), glyceryl oleate (Tegin O, GO), cetyl palmitate (CP), methylchloroisothiazolinone and methylisothiazolinone (Acnibio AC) and imidazolidinyl urea (Kemipur 100) were purchased from ACEF (Fiorenzuola d'Arda, PC, Italy). Idebenone (IDE) was a kind gift of Wyeth Lederle (Catania, Italy). All other reagents were of analytical grade and used as supplied.

### 3.2. Preparation of SLN

Unloaded and IDE-loaded SLN were prepared using the phase inversion temperature (PIT) method, as previously described [32], and their composition is reported in Table 5. The oil phase consisted of cetyl palmitate, the selected emulsifiers and different percentages (*w/w*) of IDE, while the aqueous phase was deionized water containing 0.35% *w/w* imidazolidinyl urea and 0.05% *w/w* methylchloroisothiazolinone and methylisothiazolinone as preservatives. After heating both phases separately at ~90 °C, the aqueous phase was slowly added to the oil phase, at constant temperature and under agitation. During the cooling to room temperature under slow and continuous stirring, the turbid mixture turned into clear at the PIT, whose value was recorded using a conductivity meter mod. 525 (Crison, Modena, Italy).

### 3.3. Transmission Electron Microscopy (TEM)

Transmission electron microscopy (TEM) imaging was performed using a transmission electron microscope (model JEM 2010, Jeol, Peabody, MA, USA) operating at an acceleration voltage of 200 KV. For negative-staining electron microscopy, 5 μl of sample was placed on a 200-mesh formvar copper grid (TAAB Laboratories Equipment, Berks, UK), and allowed to be adsorbed. After removing the surplus using filter paper, a drop of 2% (*w/v*) aqueous solution of uranyl acetate was added over 2 min. Then, the surplus was removed and the sample was dried at room condition prior to its imaging.

### 3.4. Photon Correlation Spectroscopy (PCS)

SLN particle sizes and polydispersity indexes (PI) were determined at room temperature using a Zetasizer Nano ZS90 (Malvern Instruments, Malvern, UK), which scattered light at 90°. The instrument performed particle sizing by means of a 4-mW laser diode operating at 670 nm. Mean diameter and PI values were the averages of results obtained for 3 replicates of 2 separate preparations.

The determination of the ζ-potential was performed using the technique of laser Doppler velocimetry using Zetasizer Nano ZS 90 after sample dilution with KCl 1 mM (pH 7.0), according to a procedure already reported [33].

### 3.5. MLV Preparation

MLV were prepared with and without IDE. Stock solutions of DMPC and IDE were prepared in chloroform/methanol (1:1, *v:v*). Aliquots of the DMPC solution were distributed in glass tubes to have the same amount of DMPC (0.010325 mmol) in each tube. Aliquots of IDE solution were added to some of these tubes containing DMPC to have a 0.06 molar fraction of IDE with respect to DMPC. The solvents were evaporated under a nitrogen stream, and the obtained films were freeze-dried to remove possible traces of solvents. At this stage, the samples were divided in two batches. The samples of one of these batches were hydrated with bi-distilled water in order to have 0.007375 mmol of DMPC in 120 μL. These samples were used to perform the calorimetric characterization of IDE loaded and unloaded MLV. The samples of the other batch were hydrated with bi-distilled water to have 0.007375 mmol of DMPC in 88.4 μL. These samples were used to carry out the interaction study of unloaded MLV (membrane model) with SLN. After the addition of bi-distilled water, all samples were kept for 1 min at 37 °C and shaken for 1 min, for 3 times, and finally kept 60 min at 37 °C.

### 3.6. Differential Scanning Calorimetry Analyses

Calorimetric analyses were performed by a STAR^e^ system (Mettler Toledo, Greifensee, Switzerland) equipped with a DSC-822^e^ calorimetric cell, using Mettler TA-STAR^e^ software. The calorimetric system was calibrated, in temperature and enthalpy changes, following the procedure of the DSC 822 Mettler TA STAR^e^ instrument, by using indium, stearic acid, and cyclohexane. In all the experiments, the reference pan was filled with the same buffer present in the sample under investigation.

Firstly, the calorimetric evaluation of IDE loaded and unloaded SLN and IDE loaded and unloaded MLV was performed, in order to obtain the corresponding reference curves. A quantity of 100 μL of SLN was transferred from the bulk to the aluminum crucible that was hermetically sealed and submitted to calorimetric analysis consisting of a heating scan from 5 to 65 °C at a scan rate of 2 °C/min and a cooling scan from 65 to 5 °C at a scan rate of 4 °C/min. These 2 scans were repeated 3 times sequentially, to check the result reproducibility.

A quantity of 120 μL of the MLV sample was put into the calorimetric pan, hermetically closed and submitted to the following cycles: a heating scan from 5 to 37 °C at a scan rate of 2 °C/min); a cooling scan from 37 to 5 °C at a scan rate of 4 °C/min; repeated at least 3 times. As all investigated thermograms were reproducible, only the third heating scan of each experiment was used to study the thermotropic behavior of the sample. Subsequently, after the calorimetric characterization of each investigated sample, the study of the interaction between SLN and MLV was carried out. A quantity of 88.4 μL of MLV was placed in the calorimetric pan and a volume of loaded SLN, chosen in order to have a 0.06 molar fraction of IDE with respect to DMPC, was added into the calorimetric pan. Bi-distilled water was added to obtain a final volume of 100 μL. The same experiments were performed with unloaded SLN (we used a volume corresponding to the same cetyl palmitate amount present in the experiments with loaded SLN).

The calorimetric pan was closed and submitted to the following scans: (i) a heating scan from 5 to 55 °C (2 °C/min); (ii) a cooling scan from 55 to 37 °C (4 °C/min); (iii) an isotherm scan of 60 min at 37 °C; (iv) a cooling scan from 37 to 5 °C (4 °C/min). This procedure was repeated at least 8 times to evaluate the interaction over time. In fact, the heating scans revealed the thermotropic behavior and changes occurring both in MLV and in SLN. The isotherm period at 37 °C allowed mimicking the physiological cell conditions, being the temperature very close to the physiological one where DMPC bilayers in MLV and the cell membrane were in a liquid crystalline phase. Each experiment was carried out in triplicate.

### 3.7. In Vitro Skin Permeation Experiments

In vitro skin permeation experiments were performed as previously reported [12,34]. Briefly, Franz-type diffusion cells with an effective diffusion area of 0.785 cm^2^ and a receiving chamber volume of 5.5 mL were used. Skin samples were excised from new-born pigs (Goland–Pietrain hybrid pigs, 1.2–1.5 kg) that died by natural causes and were provided by a local slaughterhouse. The subcutaneous fat was carefully removed and the skin was cut into squares of approximately 3 cm^2^ and stored at −80 °C. Before starting the experiments, skin samples were thawed and pre-equilibrated in physiological solution (NaCl 0.9%, *w/v*) at 25 °C for 2 h. Skin samples were placed and secured between the donor and receptor compartment of the Franz cells, with the stratum corneum (SC) facing the donor compartment. The receptor was filled with 5% Poloxamer 188 water solution, which was continuously stirred (700 rpm) and thermostated at 37 °C. A quantity of 200 µL of SLN samples was placed onto the skin surface and samples of the receiving solution were withdrawn at intervals (0, 1, 2, 4, 6, 8 and 24 h), replaced with an equal volume of 5% Poloxamer 188 water solution (pre-thermostated at 37 °C) to ensure sink conditions and analyzed by HPLC for drug content. After 24 h, the residual SLN suspension was removed from the donor compartment, the skin surface was washed and the SC was removed by stripping with adhesive tape Tesa® AG (Hamburg, Germany). The epidermis was separated from the dermis using a surgical scalpel. IDE was extracted from tape strips, epidermis, and dermis samples by immersion of the sample in methanol and subsequent sonication. The amount of IDE in the extracted samples was determined by HPLC. Each experiment was performed in triplicate. Results were expressed as cumulative amount of IDE penetrating into the different skin layers after 24 h.

HPLC analyses were carried out using a Hewlett-Packard model 1050 liquid chromatograph (Hewlett-Packard, Milan, Italy), equipped with a 20 µL Rheodyne model 7125 injection valve (Rheodyne, Cotati, CA, USA) and an UV-VIS detector (Hewlett-Packard, Milan, Italy). A Simmetry, 4.6 cm × 15 cm reverse phase column (C_18_) (Waters, Milan, Italy) was used and was eluted (flow rate 1 mL/min) with a mobile phase consisting of methanol/water 80:20, *v/v*, at room temperature. The column effluent was monitored continuously at 280 nm and IDE quantification was performed by measuring the peak areas in relation to those of a standard calibration curve obtained by plotting known concentrations of IDE vs. the corresponding peak areas. Other formulation components did not interfere with IDE quantification. The sensitivity of the HPLC method was 0.1 µg/mL.

## 4. Conclusions

In this work, we investigated the feasibility of predicting in vitro skin penetration data obtained after topical application of IDE-loaded SLN using DSC parameters, such as ΔH and T_m_, recorded during IDE-loaded SLN interactions with a model of bio-membrane consisting of MLV liposomes. We pointed out a good relationship between ΔH differences between scan 1 and scan 9 and IDE amount in the epidermis, while T_m_ values seemed to be roughly related to the amount of drug penetrating the stratum corneum. Further similar investigations are planned on SLN loaded with different drugs to corroborate the data obtained in the present study.

## Figures and Tables

**Figure 1 pharmaceuticals-11-00138-f001:**
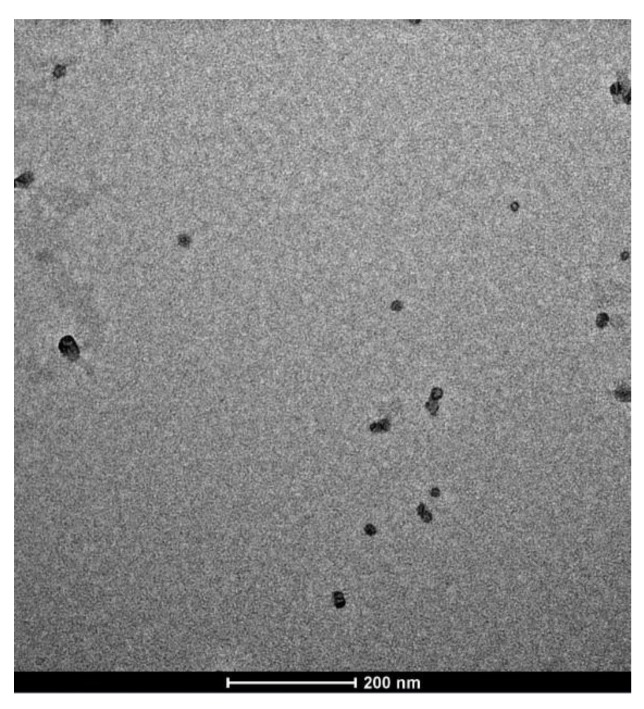
Transmission electron microscopy (TEM) image of solid lipid nanoparticles (SLN) containing oleth-20 as surfactant and idebenone (IDE) 1.1% *w/w*.

**Figure 2 pharmaceuticals-11-00138-f002:**
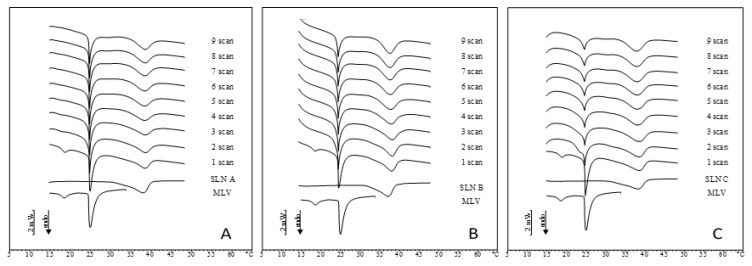
Differential scanning calorimetry (DSC) scans during the interactions of unloaded SLN with multi-lamellar (MLV) liposomes. A = SLN A; B = SLN B; C = SLN C.

**Figure 3 pharmaceuticals-11-00138-f003:**
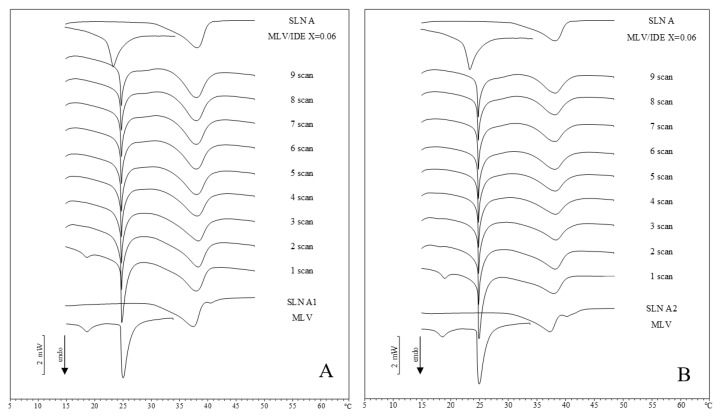
DSC scans during the interactions of IDE-loaded SLN with MLV liposomes. A = SLN A1; B = SLN A2.

**Figure 4 pharmaceuticals-11-00138-f004:**
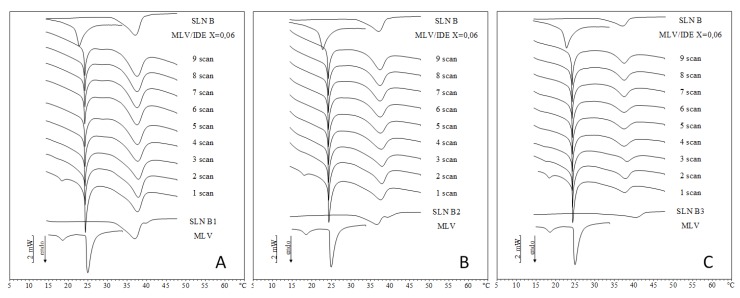
DSC scans during the interactions of IDE-loaded SLN with MLV liposomes. A = SLN B1; B = SLN B2; C = SLN B3.

**Figure 5 pharmaceuticals-11-00138-f005:**
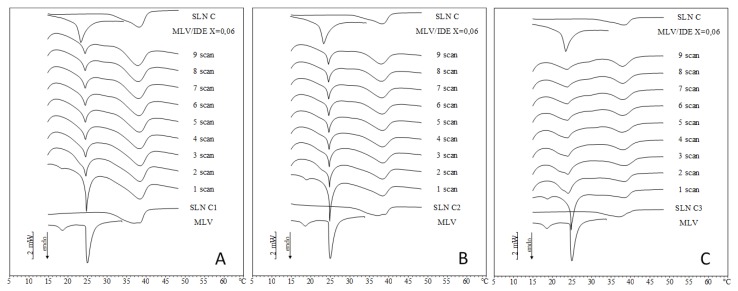
DSC scans during the interactions of IDE-loaded SLN with MLV liposomes. A = SLN C1; B = SLN C2; C = SLN C3.

**Figure 6 pharmaceuticals-11-00138-f006:**
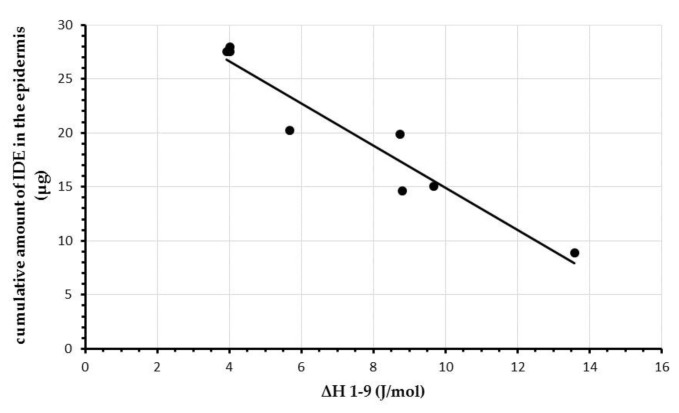
Relationship between the cumulative amounts of IDE penetrated in the epidermis and ΔH differences (ΔH1–9) between the first and the last scan. Each data represents the mean of three different experiments. Standard deviations (S.D.) were omitted for clarity and were within 5% for ΔH values.

**Figure 7 pharmaceuticals-11-00138-f007:**
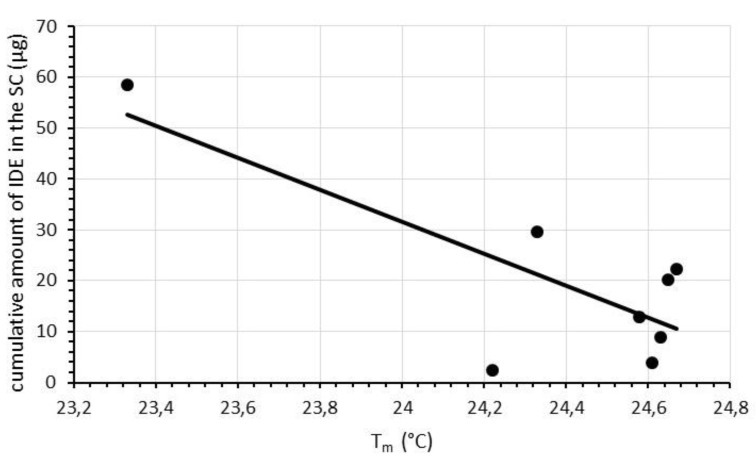
Relationship between the cumulative amounts of IDE penetrated in the stratum corneum (SC) and T_m_ values determined during scan 9 for the corresponding IDE-loaded SLN. Each data represents the mean of three different experiments. Standard deviations (S.D.) were omitted for clarity and were within 5% for T_m_ values.

**Figure 8 pharmaceuticals-11-00138-f008:**
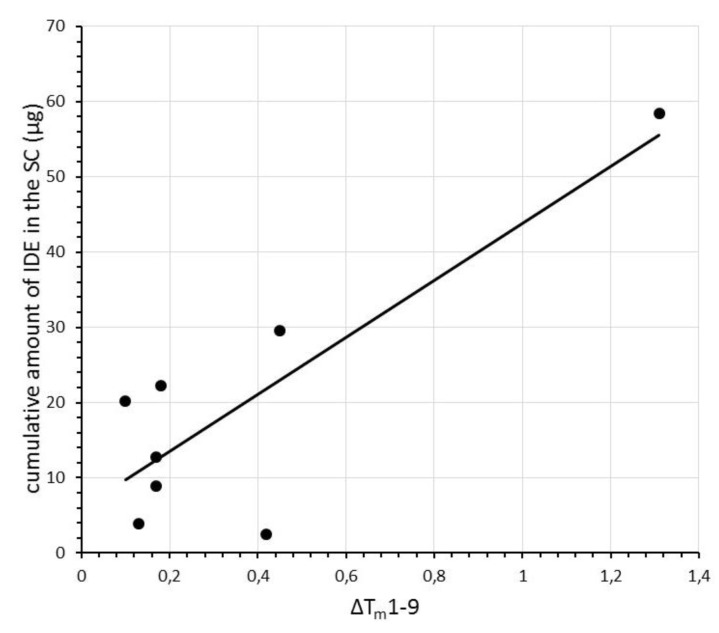
Relationship between the cumulative amounts of IDE penetrated in the stratum corneum (SC) and T_m_ differences (ΔT_m_1–9) between the first and the last scan. Each data represents the mean of three different experiments. Standard deviations (S.D.) were omitted for clarity and were within 5% for T_m_ values.

**Table 1 pharmaceuticals-11-00138-t001:** Characterization of unloaded and IDE-loaded SLN: particle size (size ± S.D.), polydispersity indexes ± S.D. (PI ± S.D.), ζ potential (± S.D.) and phase inversion temperature (PIT) values.

SLN	Size ± S.D. (nm)	PI ± S.D.	ζ potential ± S.D. (mV)	PIT (°C)
A	42.2 ± 0.8	0.268 ± 0.014	−1.76 ± 0.27	80
A1	41.7 ± 0.7	0.281 ± 0.013	−1.66 ± 0.38	80
A2	43.1 ± 0.9	0.272 ± 0.018	−2.35 ± 0.37	80
B	45.4 ± 0.6	0.281 ± 0.016	−1.90 ± 0.39	81
B1	46.6 ± 0.9	0.290 ± 0.015	−1.98 ± 0.45	80
B2	44.9 ± 1.0	0.275 ± 0.021	−2.29 ± 0.44	80
B3	33.9 ± 0.8	0.167 ± 0.016	−1.90 ± 0.58	81
C	36.1 ± 0.6	0.178 ± 0.017	−1.87 ± 0.25	85
C1	35.3 ± 0.4	0.159 ± 0.019	−1.78 ± 0.29	85
C2	36.4 ± 0.5	0.183 ± 0.022	−2.41 ± 0.58	84
C3	35.9 ± 0.7	0.155 ± 0.017	−1.94 ± 0.29	84

**Table 2 pharmaceuticals-11-00138-t002:** Cumulative amount of IDE (µg) penetrating different skin layers (SC = stratum corneum; E = epidermis; D = dermis) after 24 h from IDE-loaded SLN. Each value represents the mean of three different experiments ± S.D.

SLN	SC	E	D
A1	20.23 ± 2.84	27.95 ± 5.79	1.47 ± 0.24
A2	22.20 ± 6.25	27.54 ± 6.87	1.40 ± 0.76
B1	3.86 ± 1.36	14.57 ± 1.39	0.71 ± 0.08
B2	8.94 ± 2.57	20.21 ± 7.26	1.64 ± 0.19
B3	12.79 ± 1.76	27.51 ± 3.25	4.21 ± 0.66
C1	2.50 ± 0.59	8.90 ± 2.31	0.26 ± 0.08
C2	29.62 ± 1.19	15.08 ± 4.93	2.22 ± 0.32
C3	58.47 ± 5.76	19.87 ± 2.67	3.44 ± 0.34

**Table 3 pharmaceuticals-11-00138-t003:** Enthalpy change (ΔH) values determined for each DSC scan during in vitro experiments on IDE-loaded SLN interactions with MLV liposomes and ΔH difference between scan 1 and scan 9 (ΔH1–9).

Sample	ΔH (J/mol)									ΔH1–9 (J/mol)
	scan 1	scan 2	scan 3	scan 4	scan 5	scan 6	scan 7	scan 8	scan 9	
A1	27.06	26.38	24.12	24.58	24.11	24.04	23.57	23.10	23.04	4.02
A2	27.58	24.25	24.25	23.57	23.65	23.93	23.22	23.36	23.65	3.93
B1	29.69	29.09	26.75	23.30	23.52	22.53	21.24	20.84	20.88	8.81
B2	30.01	30.34	25.93	25.38	24.38	24.22	24.49	24.53	24.34	5.67
B3	28.40	28.39	26.42	26.12	26.50	25.50	24.61	24.11	24.39	4.01
C1	28.32	27.90	26.43	21.63	20.18	17.96	15.99	15.23	15.13	13.19
C2	27.16	26.70	24.77	21.98	20.56	19.25	18.39	17.58	17.48	9.68
C3	27.53	24.56	23.65	21.89	20.36	19.54	18.65	18.70	18.79	8.74

**Table 4 pharmaceuticals-11-00138-t004:** Transition temperature (T_m_) values determined for each DSC scan during in vitro experiments on IDE-loaded SLN interactions with MLV liposomes and ΔT_m_ difference between scan 1 and scan 9 (ΔT_m_1–9).

Sample	T_m_ (°C)									T_m_1-9 (°C)
	scan 1	scan 2	scan 3	scan 4	scan 5	scan 6	scan 7	scan 8	scan 9	
A1	24.75	24.67	24.67	24.67	24.68	24.64	24.64	24.65	24.65	0.10
A2	24.85	24.75	24.72	24.72	24.69	24.70	24.70	24.67	24.67	0.18
B1	24.74	24.67	24.64	24.64	24.61	24.61	24.61	24.61	24.61	0.13
B2	24.80	24.75	24.72	24.69	24.69	24.66	24.66	24.63	24.63	0.17
B3	24.75	24.73	24.71	24.68	24.68	24.64	24.65	24.61	24.58	0.17
C1	24.64	24.51	24.42	24.35	24.29	24.29	24.26	24.26	24.22	0.42
C2	24.78	24.65	24.59	24.53	24.46	24.43	24.40	24.36	24.33	0.45
C3	24.64	23.68	23.56	23.49	23.49	23.46	23.36	23.36	23.33	1.31

**Table 5 pharmaceuticals-11-00138-t005:** Composition of the oil phase of unloaded and IDE-loaded SLN. Values are expressed as % w/w with respect to the total amount of SLN suspension.

SLN	Isoceteth-20	Ceteth-20	Oleth-20	GO	CP	IDE
A	10.6	-	-	3.5	7.0	-
A1	10.6	-	-	3.5	7.0	0.5
A2	10.6	-	-	3.5	7.0	0.7
B	-	8.7	-	4.4	7.0	-
B1	-	8.7	-	4.4	7.0	0.5
B2	-	8.7	-	4.4	7.0	0.7
B3	-	8.7	-	4.4	7.0	1.1
C	-	-	7.5	3.7	7.0	-
C1	-	-	7.5	3.7	7.0	0.5
C2	-	-	7.5	3.7	7.0	0.7
C3	-	-	7.5	3.7	7.0	1.1

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
