# Peer review of "Differential Scanning Calorimetry Analyses of Idebenone-Loaded Solid Lipid Nanoparticles Interactions with a Model of Bio-Membrane: A Comparison with In Vitro Skin Permeation Data"

_pharmaceuticals, 2018, doi:10.3390/ph11040138_

Reviewer 1 Report

Montenegro et al., tried to establishment possibility of qualitatively predicting in vitro IDE skin penetration from IDE loaded SLN taking into DSC and test possibility of calculating IDE skin penetration using DSC analyses. The author did interactions experiments were performed using multi-lamellar liposomes as a model of bio-membrane by monitoring ΔH and Tm by DSC scans. Authors also detected that the change of ΔH and Tm values between the first and the last scan seemed to be related to SLN ability to locate IDE in the epidermis and the stratum corneum.

My minor points of this study are here 

1.    Figure 1. Must be replaced  with SLN distributions  (at least 10-15 particles)

2.    Line 94-96 “When subjected to heating, phospholipid bilayer undergoes a phase transition from gel ordered state (Lβ), where phospholipid are tightly packed, to liquid-crystalline state (Lα), in which phospholipid are more loosely associated” quote has to need appropriate citations

3.    Table 3-4. Change ΔH and Tm should be provided in heat map plotting rather than using conventional tubular format 

4.    Where are other characterization data to reconfirmation of DSC data  (XRD, TGA/DTA, etc)?

Author Response

Reviewer 1.

1.    Figure 1. Must be replaced  with SLN distributions  (at least 10-15 particles)

Answer

As requested by the reviewer, we replaced Fig.1 with a figure showing at least 10-15 nanoparticles.

2.    Line 94-96 “When subjected to heating, phospholipid bilayer undergoes a phase transition from gel ordered state (Lβ), where phospholipid are tightly packed, to liquid-crystalline state (Lα), in which phospholipid are more loosely associated” quote has to need appropriate citations

Answer

To comply with the reviewer’s request, we added the following reference:

London E, Brown DA. Insolubility of lipids in triton X-100: physical origin and relationship to sphingolipid/cholesterol membrane domains (rafts). Biochim Biophys Acta. 2000, 1508, 182-95. DOI:10.1016/S0304-4157(00)00007-1

We changed the numbering of the subsequent references accordingly.

3.    Table 3-4. Change ΔH and Tm should be provided in heat map plotting rather than using conventional tubular format.

Answer

We thank the reviewer for the suggestion. We are aware that figures have a better visual impact than tables. As stated in the introduction, the aim of this manuscript was to investigate the possibility of correlating in vitro skin penetration results observed for IDE loaded SLN with DSC data obtained studying the interactions of these SLN with a model of bio-membrane consisting of MLV liposomes. Of course, we could not report all the figures showing the correlation between the calorimetric parameters of each formulation and the corresponding skin penetration results. Therefore, we decided to report the results of DSC analyses and in vitro skin penetration results in tables to provide the reader with the opportunity of verifying the existence of a possible relationship between calorimetric data and skin penetration results. Reporting the data in figures would prevent the reader from knowing the exact values of each data, as many numbers are very close and would overlap in a graphical representation. 

4.    Where are other characterization data to reconfirmation of DSC data (XRD, TGA/DTA, etc)?

Answer

As clearly stated in the manuscript, we have already reported SLN with identical composition in previous papers published in renowned international journals. These nanoparticles were characterized by DLS, TEM and DSC and all collected data clearly demonstrated that these nanoparticles were SLN. Therefore, we did not perform any further study to characterize these SLN. In addition, the techniques mentioned by the reviewer (XRD, TGA/DTA etc.) could not provide any information about the interaction between SLN and a model of biological membrane consisting of MLV liposomes.

Reviewer 2 Report

In this manuscript, the authors described DSC analyses of idebenone-loaded SLN and interactions with a model of bio-membrane.. The results are interesting. However, this study has a few major limitations. 1. Overall, moderate English changes required. 2. The quality of TEM image is poor and the shape of SLN is not spherical. 3. Figure 2-5 are not clear. 4. In this manuscript, table 2 is one of most important results. The authors should provide figure instead of putting only table. 5. The method of penetration is missing.

Author Response

Reviewer 2.

In this manuscript, the authors described DSC analyses of idebenone-loaded SLN and interactions with a model of bio-membrane. The results are interesting. However, this study has a few major limitations.

1. Overall, moderate English changes required.

Answer

To comply with the reviewer’s request, a native English speaker has revised the manuscript.

2. The quality of TEM image is poor and the shape of SLN is not spherical.

Answer

The TEM image has been replaced. We agree with the reviewer that the shape of SLN is not perfectly spherical. Therefore, we changed the text (line 61 revised version) as follows: Roughly round-shaped nanoparticles (Fig. 1)…

3. Figure 2-5 are not clear.

Answer

We adapted Fig. 2-5 to fit the page width to allow an easy comparison among the thermograns obtained for the same type of SLN containing different percentages of IDE or among the different unloaded SLN. In our opinion, it is important for the reader to see together all the panels of the same figure. Bigger figures would have requested a different positioning and would require the reader to print or to make the text smaller to compare the different panels of each figure. Fig. 2-5 originate from the software of the DSC equipment and they have the best resolution we could obtain translating them into the manuscript template. 

4. In this manuscript, table 2 is one of most important results. The authors should provide figure instead of putting only table.

Answer

We thank the reviewer for the suggestion. We are aware that figures have a better visual impact than tables. As stated in the introduction, the aim of this manuscript was to investigate the possibility of correlating in vitro skin penetration results observed for IDE loaded SLN with DSC data obtained studying the interactions of these SLN with a model of bio-membrane consisting of MLV liposomes. Of course, we could not report all the figures showing the correlation between the calorimetric parameters of each formulation and the corresponding skin penetration results. Therefore, we decided to report the results of DSC analyses and in vitro skin penetration results in tables to provide the reader with the opportunity of verifying the existence of a possible relationship between calorimetric data and skin penetration results. Reporting the data in figures would prevent the reader from knowing the exact values of each data. In addition, we could not report all skin penetration results in a single figure but we had to make at least three bar charts. Therefore, to be as concise as possible, we collected all skin penetration results in a single table.

5. The method of penetration is missing.

Answer

We did not describe the experimental procedures of in vitro skin permeation studies because we used the same experimental protocol reported in previous papers. However, we agree with the reviewer that a description of these experimental procedures could be useful for the reader. Therefore, we added in the materials and methods section the following paragraph illustrating the experimental protocol of in vitro skin permeation studies. (see revised version line 397):

In vitro skin permeation experiments were performed as previously reported [12, 34]. Briefly, Franz-type diffusion cells with an effective diffusion area of 0.785 cm2 and a receiving chamber volume of 5.5 ml were used. Skin samples were excised from new- born pigs (Goland–Pietrain hybrid pigs, 1.2–1.5 kg) died by natural causes that were provided by a local slaughterhouse. The subcutaneous fat was carefully removed and the skin was cut into squares of approximately 3 cm2 and stored at −80 ◦C. Before starting the experiments, skin samples were thawed and pre-equilibrated in physiological solution (NaCl 0.9%, w/v) at 25 ◦C for 2 h.  Skin samples were placed and secured between the donor and receptor compartment of the Franz cells, with the stratum corneum (SC) facing the donor compartment. The receptor was filled with 5% Poloxamer 188 water solution, which was continuously stirred (700 rpm) and thermostated at 37°C. 200 µl of SLN samples was placed onto the skin surface and samples of the receiving solution were withdrawn at intervals (0, 1, 2, 4, 6, 8 and 24 h), replaced with an equal volume of 5% Poloxamer 188 water solution (pre-thermostated at 37°C) to ensure sink conditions and analyzed by HPLC for drug content. After 24 h, the residual SLN suspension was removed from the donor compartment, the skin surface was washed and the SC was removed by stripping with adhesive tape Tesa® AG (Hamburg, Germany). The epidermis was separated from the dermis using a surgical scalpel. IDE was extracted from tape strips, epidermis, and dermis samples by immersion of the sample in methanol and subsequent sonication. The amount of IDE in the extracted samples was determined by HPLC. Each experiment was performed in triplicate. Results were expressed as cumulative amount of IDE penetrated into the different skin layers after 24 h.

HPLC analyses were carried out using a Hewlett-Packard model 1050 liquid chromatograph (Hewlett-Packard, Milan, Italy), equipped with a 20 l Rheodyne model 7125 injection valve (Rheodyne, Cotati, CA, USA) and an UV-VIS detector (Hewlett-Packard, Milan, Italy). A Simmetry, 4.6 cm × 15 cm reverse phase column (C18) (Waters, Milan, Italy) was used and it was eluted (flow rate 1 ml/min) with a mobile phase consisting of methanol/water 80:20, v/v, at room temperature. The column effluent was monitored continuously at 280 nm and IDE quantification was performed by measuring the peak areas in relation to those of a standard calibration curve obtained plotting known concentrations of IDE vs the corresponding peak areas. Other formulation components did not interfere with IDE quantification. The sensitivity of the HPLC method was 0.1 µg/ml.

Round  2

Reviewer 2 Report

I recommend a publication of this manuscript in Pharmaceuticals.

The authors have provided and addressed all the information and my points.